# Comparison of the Vibration Damping of the Wood Species Used for the Body of an Electric Guitar on the Vibration Response of Open-Strings

**DOI:** 10.3390/ma14185281

**Published:** 2021-09-14

**Authors:** Tony Ray, Jasmin Kaljun, Aleš Straže

**Affiliations:** 1Faculty of Mechanical Engineering, University of Maribor, Smetanova 17, 2000 Maribor, Slovenia; raytonymusic@gmail.com (T.R.); jasmin.kaljun@um.si (J.K.); 2Biotechnical Faculty, University of Ljubljana, Jamnikarjeva 101, 1000 Ljubljana, Slovenia

**Keywords:** electric guitar, solid body, wood, acoustic properties, vibration damping

## Abstract

Research show that the vibrations of the strings and the radiated sound of the solid body electric guitar depend on the vibrational behavior of its structure in addition to the extended electronic chain. In this regard, most studies focused on the vibro-mechanical properties of the neck of the electric guitar and neglected the coupling of the vibrating strings with the neck and the solid body of the instrument. Therefore, the aim of the study was to understand how the material properties of the solid body could affect the stiffness and vibration damping of the whole instrument when comparing ash (*Fraxinus excelsior* L.) and walnut (*Juglans regia* L.) wood. In the electric guitar with identical components, higher modal frequencies were confirmed in the structure of the instrument when the solid body was made of the stiffer ash wood. The use of ash wood for the solid body of the instrument due to coupling effect resulted in a beneficial reduction in the vibration damping of the neck of the guitar. The positive effect of the low damping of the solid body of the electric guitar made of ash wood was also confirmed in the vibration of the open strings. In the specific case of free-free vibration mode, the decay time was longer for higher harmonics of the E_2_, A_2_ and D_3_ strings.

## 1. Introduction

The solid-body electric guitar has a thick and solid wood plate with theoretically low admittance at the bridge to better support the vibration of the strings than hollow-body acoustic or electroacoustic guitars [1,2,3]. The instrument is typically equipped with electromagnetic pickups that convert the mechanical string vibration into an amplified electrical signal that is used to radiate the sound through speakers.

Unlike an acoustic guitar, the electric guitar itself emits very little sound, so there is generally no need to transfer the energy of the vibrating strings to the body of the instrument. This means that the vibrations of the strings of a solid-body electric guitar do not decay as quickly as those of an acoustic guitar [4]. Guitarists explain it this way: An electric guitar has better sustain than an acoustic guitar [5].

Studies on the acoustics of the electric guitar have so far focused mainly on the chain from the pickup to the amplifier, including effects devices for sound synthesis and post-processing of the output signal or musical analysis [2,3,6,7,8]. However, in reality, the vibrations of the strings of a solid body electric guitar are also affected by its end supports as well as by their mechanical properties, the surrounding media, the vibrations of the adjacent strings, and the vibrations of the whole instrument [9]. Recently, attention has been focused on the study of the harmonic content of standing waves in guitar strings and on the body–vibration coupling of the instrument [10,11]. However, in the coupled vibration of the body and strings of the electric guitar, it has been found that the coupling of the neck to the vibrating strings is of high importance [1,3,4,12]. Vibrations in the neck have been shown to cause dead spots at certain fret positions [13].

The body vibrations of an electric guitar are generally not examined because the string vibrations are converted to sound through the use of a magnetic pickup. However, in the general case of stringed instruments, the studies show that when the frequencies of the separated systems are close to each other, the frequencies and damping of the coupled system may change [9,14,15,16]. It was found that the mode shapes and frequencies of the solid body guitars, similar to acoustic instruments, also depend on the body shape [17,18]. It was found out, that mode shapes and frequencies are considerably different for the body, though neck vibrations are more closely related. Recently, mode shapes of electric guitars have also been studied with FEM models [3,18,19].

To the authors’ knowledge, the properties of the material used for the solid body of the electric guitar, as opposed to the studies of the materials of the neck of the guitar [20,21], are less researched. Compared to acoustic guitars, body of electric guitar is made of woods with greater density. Some guitars use laminates (e.g., plywood) or pieces of wood glued together [22]. Some reports, based on psychoacoustics and perceived tone analysis, more sustain attributed to solid body electric guitars made of heavier wood, or wood from tropical species [23].

An influence on the sound of the electric guitar is also attributed by some musicians to the solid body of the instrument and the choice of appropriate wood. Research focusing on this part of the musical instrument is rare. This study therefore aims to investigate whether the material properties of the solid body of an electric guitar have a significant influence on the vibration characteristics of this instrument and its acoustic behavior.

## 2. Materials and Methods

### 2.1. Construction and Assembling of Electric Guitar

We have made two identical bodies for the electric guitar—the first from the wood of the ash (A) (*Fraxinus excelsior* L.) and the second from the wood of the walnut (W) (*Juglans regia* L.). In both cases, we made an element with dimensions 430 mm × 186 mm × 42.8 mm (length (L) × width (R) × thickness (T)). The wood was previously conditioned, and the initial acoustic properties were determined in advance by ultrasonic measurements and by analysis of the frequency response during free flexural vibration (Table 1) [24]. The mechanical processing of the body, for the purpose of installing electric guitar components, such as neck, pickup and bridge, was carried out on the basis of a CAD model using a CNC machine (Figure 1). We have examined only one body of a single species of wood. However, in a previous larger sample, we confirmed statistically significant differences between wood species in density (*ρ_A_* = 807 kg/m^3^ (CV% = 9.3); *ρ_W_* = 623 kg/m^3^ (CV% = 7.5)), stiffness, vibration damping and other acoustic parameters (Table 1) [24]. In order to simplify the future process of building a numerical model, as part of a broader study of electric guitar design based on the acoustic properties of the material, we have included only the basic shape, i.e., a case study, of the guitar body in the initial investigation.

The same, commercially available, 62.9 cm long composite neck made of hard maple (*Acer saccharum* L.) with a fingerboard, also made of maple, was mounted on each guitar body tested. The wooden neck was equally bolted to both guitar bodies (Figure 2). We used a tension mechanism Duesenberg Z—Standard, bridge Gotoh and pickups DiMarzio DP103CR 36th Anniversary, PAF (near the neck) and DiMarzio DP223BC PAF 36th Anniversary (near the bridge). The single guitar was then equipped with steel strings (Ernie Ball 10–46), made from nickel plated wire wrapped around tin plated hex shaped steel core wire (Gauges 0.010, 0.013, 0.017, 0.026, 0.036, 0.046), and tuned to standard: E_2_ (82.4 Hz), A_2_ (110.0 Hz), D_3_ (146.8 Hz), G_3_ (196.0 Hz), B_3_ (246.9 Hz) and E_4_ (329.6 Hz).

From a mechanical point of view, these guitars tested are nominally identical, with the only difference being in the directional mechanical stiffness, i.e., the modulus of elasticity, torsion moduli and damping of the material used for the guitar body (Table 2) [24]. For both guitar bodies, the wood grain was aligned along the guitar structure (*x*-axis), the radial wood direction along the width (*x*-axis), and the tangential direction along the thickness of the guitar body (*z*-axis). The headstock and body of the guitars are practically symmetrical with respect to the longest *x*-axis, which is not typical of a conventional guitar body (Figure 2). With the same guitar equipment and identical tests, despite the coupling effect of the components of a single guitar, it is to be expected that the differences in the acoustic behavior of the tested guitars are mainly due to the material properties of the guitar body used.

### 2.2. Analysis of Acoustic Behavior of Electric Guitar

#### 2.2.1. Analysis of the Acoustic Response of Guitar at Impulse Excitation

Since string/structure coupling occurs mainly at the neck [1,25], the driving- and one of the measuring points for the flexural vibration of the guitar was taken near the nut, at the position of the eighth fret (Figure 2). The short elastic impulse excitation with the impact hammer (PCB 086C02) was performed at the selected location on the neck. During the experiment, the guitar was placed on elastic supports placed under the body of the guitar, at the location of the pickup near the neck (NP) and under the bridge. During the experiment, the guitar was placed on elastic supports placed under the body of the guitar, at the location of the pickup near the neck (NP) and under the bridge. The acoustic response of the guitar was recorded at the same position, from the back of the neck, using an accelerometer (NA; PCB 352C33). The vibration response of the guitar body was obtained using a pickup (BP; DiMarzio DP223BC PAF 36th Anniversary) and two microphones, near the pickup (M_1_; PCB130D20) and near the edge of the body (M_2_; PCB130D20). The USB NI 9234 data acquisition module, in 51.2 kHz sampling mode, was used to record the signals, which were analyzed using LabView software (National Instruments, Austin, USA). For each guitar, modal frequencies and damping ratios determined by logarithmic decrement in moving 100 ms time interval of signals are identified in the low-frequency range.

In order to achieve the maximum amplitude of vibration of the guitar neck, avoid antinodes and facilitate the transfer of energy to the guitar body, the excitation point (EP) was placed halfway along the neck (8th fret of the fingerboard). Since hammer excitation force has a smooth spectrum in the frequency bandwidth of the study, the measured signals can be said to present the impulse response of the structure. Assuming, that the excitation force is small enough to stay in a linear approximation, and using the concept of additive synthesis, the acquired signals (NA, BP, M_1_, M_2_) were considered as a sum of exponentially damped sinusoids [25,26] (Equation (1)):(1)$$s\left(t\right)=\overset{\infty }{\underset{n=1}{\sum }}\beta_{n}\sin\left(2\pi f_{n}t+\emptyset_{n}\right)e^{-\alpha_{n}t}$$
where *s* is the radiated signal as a function of time *t*, *f_n_* is the resonance frequency of order *n*, and *Φ_n_* is the phase shift. A bandpass filter near the resonance frequencies (*f_n_* ± 0.1 *f_n_*) was used to simultaneously determine resonance frequency *f_n_*, the amplitude *β_n_*, and the temporal damping *α_n_*, which was determined from the FFT analysis. Only the first three resonant modes (*n* ≤ 3) were considered due to their high energy in order to keep the error for *f_n_*, *β_n_* and *α_n_* below 0.1%.

The combined use of waveguide synthesis and additive synthesis model allowed the calculation of the internal friction *tan*δ, i.e., the damping coefficient of the signal at each resonance frequency (Equation (2)). The damping coefficient *tan*δ is related to the concept of complex modulus and represents the ratio between the modulus of viscosity and the modulus of elasticity of the tested material [1,26].
(2)$$tan\delta =\frac{\alpha_{n}}{\pi f_{n}}$$

#### 2.2.2. Analysis of Vibration of Open Strings in Time and Frequency Domain

Each string was excited by hand with a pick by an experienced guitarist. The excitation point was located between two pickups, i.e., the body pickup and the neck pickup, at the typical playing distance of 12 cm from the bridge (Figure 2). The excitation angle between the *y*- and *z*-axes of the string was 45° into the body to excite both string polarizations identically. To ensure repeatability, each string was excited six times, the individual signals were analyzed and data variability was evaluated using descriptive statistics. Comparison of data obtained in repeated experiments showed that the reproducibility was satisfactory.

The radiated sound signals were captured similarly to the impulse excitation of the electric guitar by the body pickup (BP; DiMarzio DP2238C-PAF) and the USB NI 9234 data acquisition module with a sampling rate of 51.2 kHz (*t_capture_* = 7 s). We used the concept of additive synthesis (Equation (1)), extracting the first seven modal frequencies (*n* ≤ 7) for vibrating strings using FFT. For the analysis of the decay rate of the different string harmonics, the bandpass filter around the individual resonance frequency (*f_n_* ± 0.1 *f_n_*) was used, with the damping coefficient determined by Equation (2). In addition, the visualization of the time–frequency sonograms was performed by Short-Time Fourier Transformation (STFT) to qualitatively analyze the behavior of the FFT spectra in the recorded 7 s long time scale.

## 3. Results

### 3.1. Acoustic Response of Electrical Guitar at Mechanical Excitation

For each measurement on both guitars the identification of modal parameters is based on signal from accelerometer, mounted on the neck (NA), signal from the body pickup (BP), and signals from two microphones (M_1_, M_2_). In the investigated low-frequency interval (<500 Hz), the modal frequencies determined from a single measurement signal on the electric guitar with the body of ash wood were on average 118.0 Hz in the 1st vibration mode and 203.1 Hz and 438.0 Hz in the 2nd and 3rd vibration modes (Figure 3, Table 3). At the selected measurement points of the electric guitar with walnut body, the modal frequencies averaged 108.2 Hz in the 1st vibration mode and 200.6 Hz and 419.2 Hz in the 2nd and 3rd vibration modes.

The highest vibration damping was measured at the body of both guitars in the 1st vibration mode (*tan*δ_A_ = 0.093, *tan*δ_W_ = 0.121), lower in the 2nd vibration mode (*tan*δ_A_ = 0.053, *tan*δ_W_ = 0.073) and the lowest in the 3rd vibration mode (*tan*δ_A_ = 0.022, *tan*δ_W_ = 0.026) (Figure 4). The damping on the walnut body was statistically significantly greater than that of a guitar with the body of ash wood. The measured damping of the signal by the pickup (BP) was of the same size for all investigated modal frequencies as for the measurements on the body with the microphone. With a pickup, statistical significantly higher signal attenuation were determined at guitar with body from walnut in the 1st vibration mode (*tan*δ_A_ = 0.114, *tan*δ_W_ = 0.119) and in the 3rd vibration mode (*tan*δ_A_ = 0.026, *tan*δ_W_ = 0.046), while we measured similar values in the 2nd vibration mode for both guitars (*tan*δ_A_ = 0.072, *tan*δ_W_ = 0.073).

The differences in the absolute values of vibration damping are due to the different sensing principle of the devices and their positions (Figure 4). The condenser microphone (M_1_) uses the indirect mechanical principle and the guitar pickup (BP) the electromagnetic principle. However, the correlation of the vibration damping detected by both devices is confirmed, but only in the 1st mode. We hypothesize that the correlation is not characteristic in higher vibration modes, which is due to the increasing influence of the distance between the two sensors (Figure 5).

We found that some modal frequencies occur at all the studied positions of the electric guitar and differ only insignificantly from each other (Table 1, Figure 3). This allowed verifying the existence of the coupling effect of vibrations of individual parts of the guitar. Thus, when testing the correlation between the damping of the guitar body (M1) and the guitar neck (NA), we confirmed the statistical significance in both guitars, with the stronger correlation occurring at the modal frequency of the 1st and 3rd vibration modes. It seems that the damping at the neck of a guitar with a walnut body is greater and is due to the greater vibration damping of the source material itself used for the body of this guitar. However, the finding seems to be specific, and the measured damping could also depend on the vibration mode or on the location and type of the measurement sensor (Figure 5).

We also checked whether there is a correlation between the damping of the mechanical vibration of the guitar body (M_1_) and the signal to the pickup (BP), which detects the vibration of the strings above, when the strings are indirectly excited by the vibration of the whole structure of the guitar. In this case, the correlation is statistically significant, but only at the 1st modal frequency and, for the guitar with the body of an ash wood, also at the 2nd modal frequency (Figure 5). The correlation could not be confirmed for the damping in the higher vibration modes of both guitars.

### 3.2. Acoustic Response of Electric Guitar at Vibration of Open Strings

The resulting data set on the vibration of open strings is quite extensive, and it is not easy to convey the full range of information. Decay times did not differ significantly when comparing the two guitars or between individual excited strings in the fundamental frequency mode (Figure 6 and Figure 7). At higher harmonics, the decay rate, i.e., the damping coefficient, was found to be significantly different only for the vibration of the E_2_ and A_2_ strings, starting from the 2nd harmonic for the E_2_ string and from the 3rd harmonic for the A_2_ string (Figure 6).

For thinner strings (G_3_, B_3_ and E_4_), the damping coefficient was generally highest at the fundamental frequency and decreased significantly with increasing frequency. The study confirmed only minor differences in the damping coefficient between the guitars tested in the case of the G_3_, B_3_ and E_4_ strings (Figure 7). Greater damping was found only for the 1st vibration mode of the B_3_ string in the case of the guitar with walnut body. For the higher vibration modes, the vibration damping was generally lower but different, with no discernible difference between the wood species tested.

The confirmed greater damping of open string vibration at higher modes, in the case of the E_2_, A_2_ and partial D_3_ string (3 < *n* ≤ 7) in the walnut-bodied guitar, was further confirmed by STFT time–frequency spectrograms (Figure 8). The differences in the amplitude of the vibrations in this case were detected in the relative color scale, from low intensities determined by blue to high intensities determined by red. The color mapping of the spectra of the G_3_, B_3_ and E_4_ string signals showed no discernible differences in the signals between the two guitars tested.

## 4. Discussion

When we made the solid body of an electric guitar from two types of wood that have different acoustic–mechanical properties, we found that they have some influence on the vibro-acoustic properties of the guitar body as well as the whole instrument. As this study has shown, by using a material, i.e., ash wood with greater mechanical stiffness (Table 2), we achieve higher modal frequencies of the body of the electric guitar as well as of the whole instrument (Table 3). In this way, it also influences the characteristics, i.e., the modal frequency of the guitar neck, which is otherwise considered to be more important for the acoustic behavior of an electric guitar compared to the body of this instrument [3,13,27].

It can be said that the mechanical admittance of an electric guitar neck, parallel and perpendicular to the fretboard, is still largely in the domain of its construction and material selection, as established by previous studies [3,4,13,20]. Regardless, the part of the mechanical admittance of the guitar neck, the real part of which is called the conductance and represents the flow of energy from the vibrating strings to the neck, can also be attributed to the vibromechanical properties of the guitar body. This was also confirmed in this study by the characteristic correlation between the damping of the mechanical vibration of the guitar neck and the damping measured on the body of the electric guitar when mechanically excited (Figure 5). It is shown that in an electric guitar it is necessary to choose a wood with low damping properties for the body as well as for the neck in order to achieve a sufficiently small damping of the mechanical vibration of the whole system. In this respect, ash wood has proven to be more suitable, due to generally higher mechanical stiffness and lower vibration damping (Table 2). Otherwise, we recommend heavier woods with a more ordered anatomical structure, which generally have a low damping of mechanical vibration [23,28,29,30].

This study also confirmed that at closely spaced frequencies of separated systems, the frequencies and the damping of the coupled system can be altered [16,25]. At higher vibration modes of strings E_2_, A_2_ and partially D_3_, we found a characteristic influence of the damping properties of the guitar body, at the vibration frequencies of the strings close to the modal frequencies of the guitar. A longer decay time in higher vibration modes was confirmed in this case for a guitar with a body made of ash wood, which has a lower damping of mechanical vibrations. This finding is limited to the free vibration of the strings of the electric guitars tested under laboratory conditions in so called open strings vibration mode, having the tested guitar on elastic supports.

This research is part of a project to develop a design protocol for electric guitars that allows the designer to anticipate and, if necessary, influence the final sound of the guitar during the virtual development phase. Under the particular laboratory conditions, the results show that the material properties of the guitar body indeed have an impact on the vibrations of the whole guitar structure. This will be further investigated by creating a numerical model and validating it.

Since we know that the boundary conditions for playing the electric guitar in practice are very different, it is necessary for a comprehensive analysis and confirmation of these results to also further investigate the function of this instrument under real conditions. In this way, these results can be better understood and compared with previous studies, especially for study cases in which the instrument is played under real environmental conditions [3,4,13,14].

## 5. Conclusions

This study confirms that the elastomechanical and acoustic properties of the material used for the solid body of an electric guitar also affect the acoustic properties of the instrument. In the specific case of the free-free vibration mode, the correlation of the vibration damping of the solid body and the neck of the guitar is characteristic. Under these conditions, the research confirmed the advantages of using stiffer ash wood, which has lower damping compared to walnut wood, for the solid body of the electric guitar when the other identical components are used. In fact, choosing walnut with greater vibration damping for the guitar body has the negative effect of shortening the decay time of some open vibrating strings due to the coupling effect with the guitar neck and strings.

## Figures and Tables

**Figure 1 materials-14-05281-f001:**
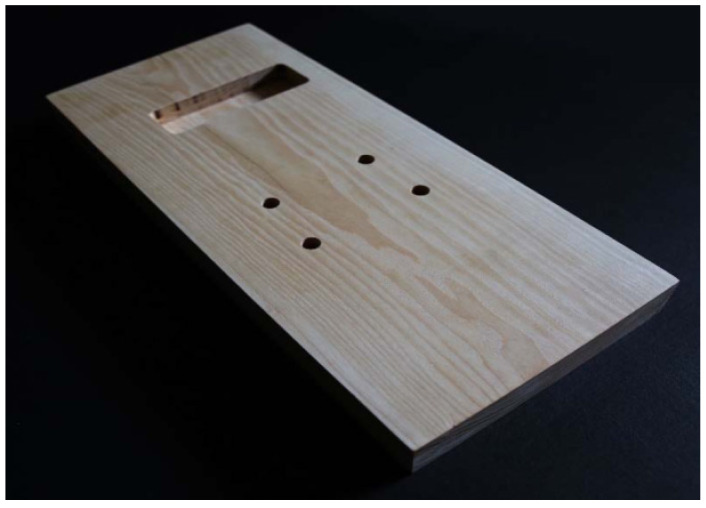
CNC-machined element of ash (A) (*Fraxinus excelsior* L.) for the solid body of an electric guitar with holes for pickup, bridge and tailpiece [24].

**Figure 2 materials-14-05281-f002:**
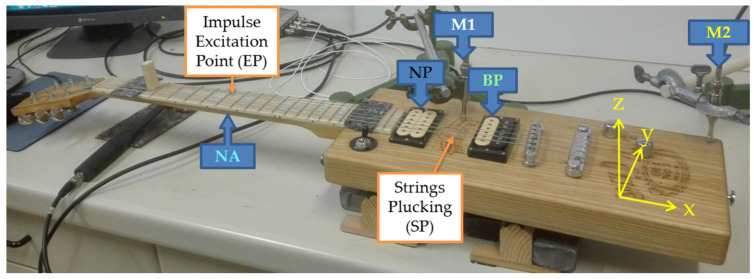
Test stand scheme for analysis of electric guitar behavior (guitar with body from walnut): EP—point of short, elastic impact excitation; SP—strings plucking, NA—position of neck accelerometer, NP—neck pickup, BP—body pickup, M_1_—body microphone, M_2_—edge body microphone.

**Figure 3 materials-14-05281-f003:**
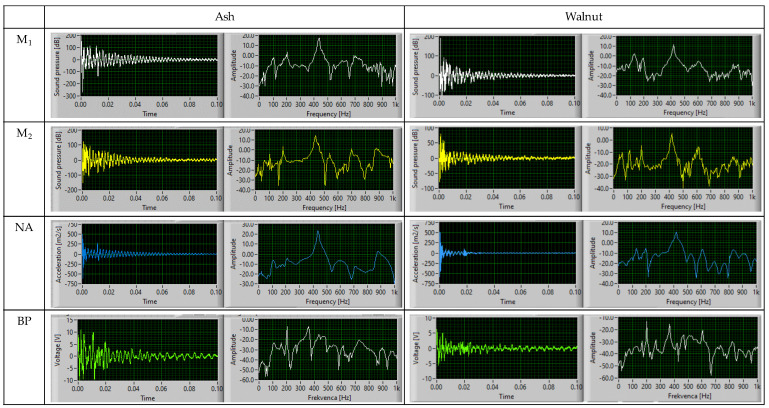
Acquired signals in time (**left**) and frequency domain (**right**) on tested positions of guitars after impulse mechanical excitation (M_1_—body microphone; M_2—_edge body microphone; NA—8th fret of fingerboard; BP—body pickup) with body made of ash and walnut wood.

**Figure 4 materials-14-05281-f004:**
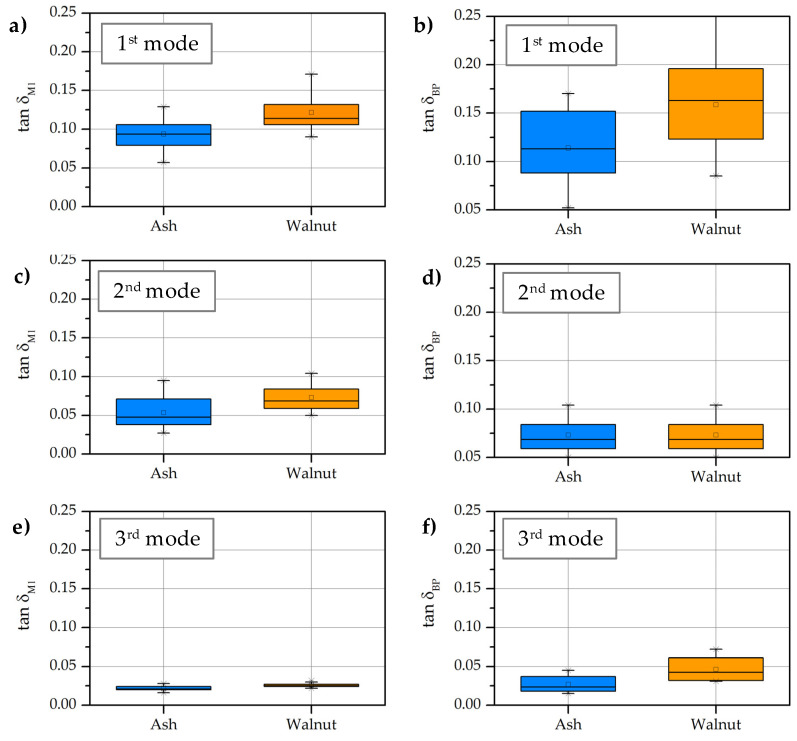
Vibration damping of guitar body made from ash and walnut wood in 1st, 2nd and 3rd vibration mode: (**a**,**c**,**e**)—signal acquisition with body microphone (M_1_); (**b**,**d**,**f**)—signal acquisition with body pickup (BP) [(ash wood (**●**); walnut wood (**●**)].

**Figure 5 materials-14-05281-f005:**
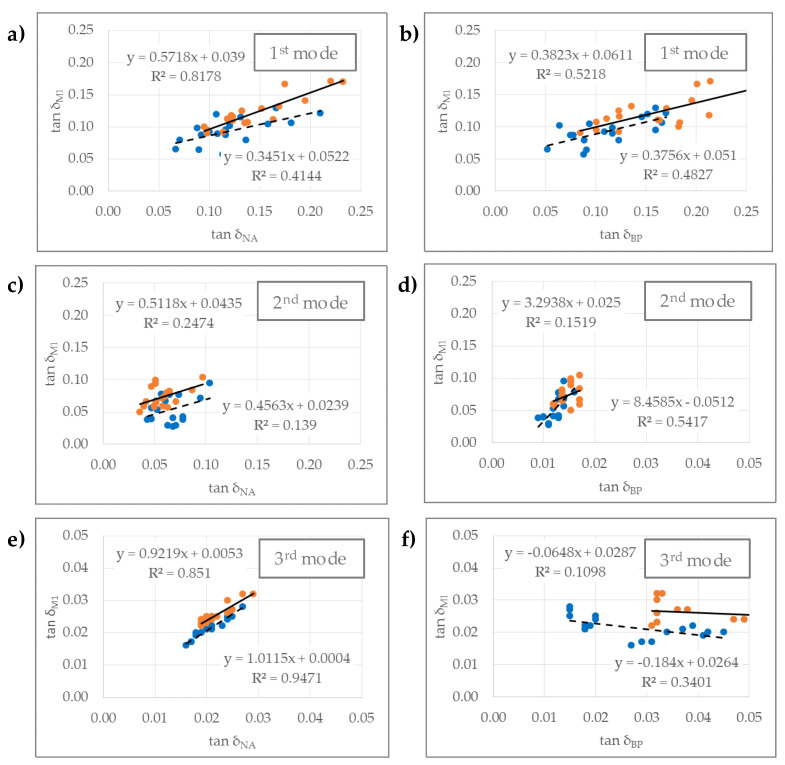
Correlation of damping of mechanical vibrations at the neck (NA, 8th fret) and body of the guitar (M_1_; (**a**,**c**,**e**)), and at the body of the guitar (M_1_) and the pickup (BP; (**b**,**d**,**f**)) [(ash wood (**●**); walnut wood (**●**)].

**Figure 6 materials-14-05281-f006:**
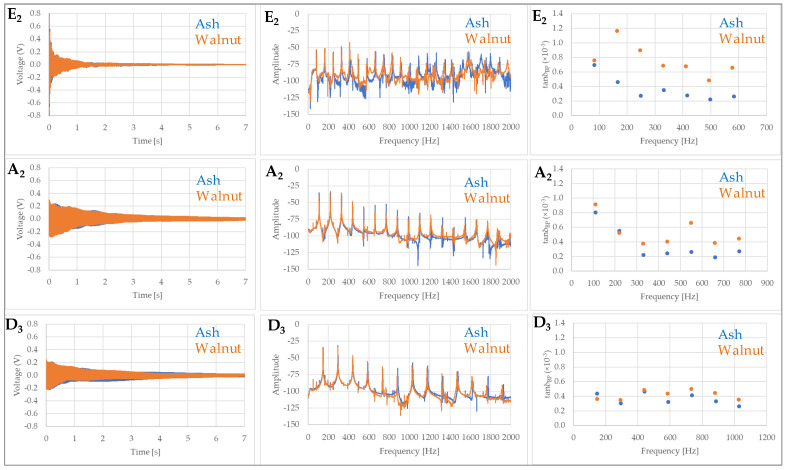
Measured wavelet–time signal, FFT frequency spectra and damping coefficient in the first 7 vibration modes of the open strings (E_2_—top; A_2_—middle, D_3_—bottom) of an electric guitar with body made of ash wood (**―**) and walnut wood (**―**).

**Figure 7 materials-14-05281-f007:**
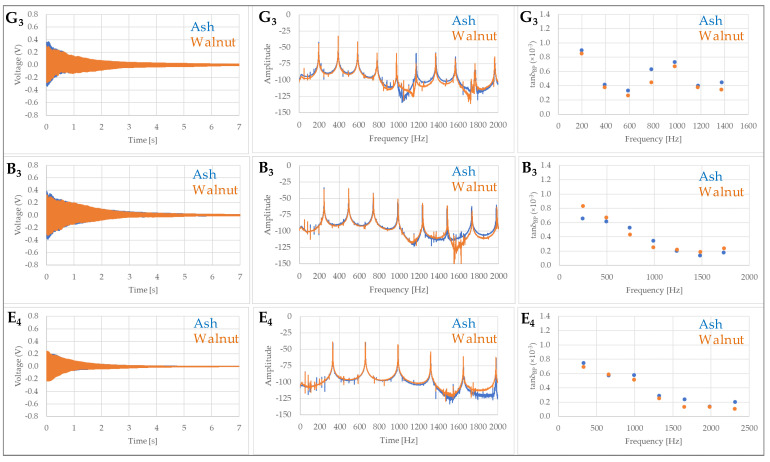
Measured wavelet time signal, FFT frequency spectra and damping coefficient in the first 7 vibration modes of the open strings (G_3_—top; B_3_—middle, E_4_—bottom) of an electric guitar with body made of ash wood (**―**) and walnut wood (**―**).

**Figure 8 materials-14-05281-f008:**
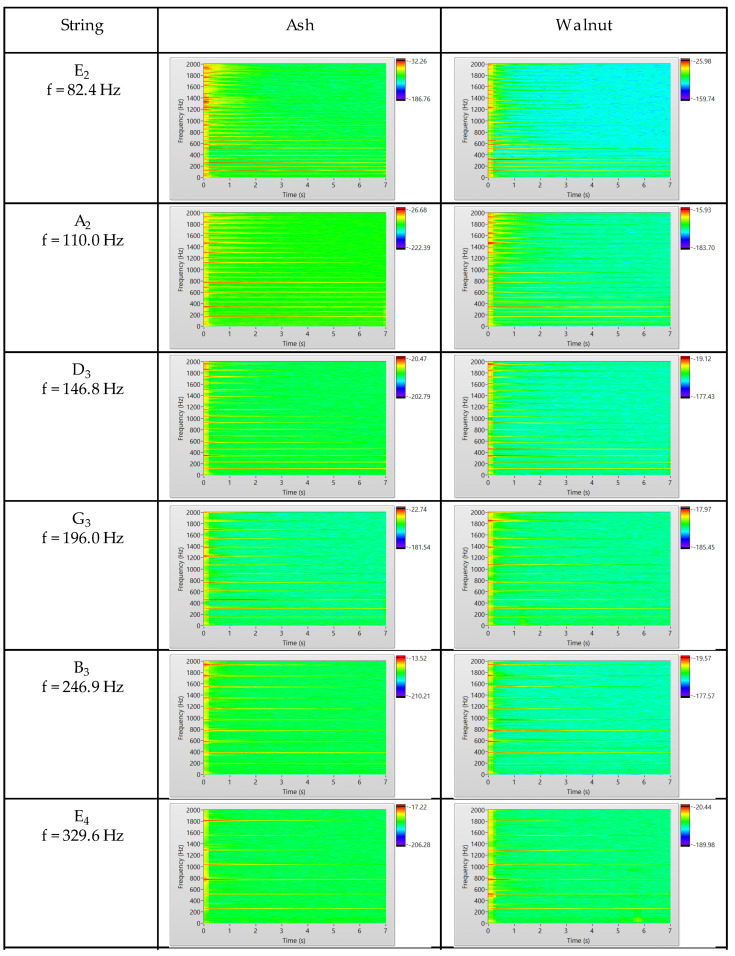
Short-Time Fourier Transformation (STFT) spectrograms of the vibrating strings (E_2_, A_2_, D_3_, G_3_, B_3_ and E_4_) of the guitar with the body made of ash wood (left) and with the body made of walnut wood (right).

**Table 1 materials-14-05281-t001:** Mean acoustic quality indicators of ash (A) and walnut (W) wood, determined by flexural vibration of elements for electric guitar solid body: *f*_1_—fundamental frequency, *tanδ*—vibration damping, *E*/*ρ*—specific modulus of elasticity, *K*—acoustic coefficient, *ACE*—acoustic conversion efficiency and *RACE*—relative acoustic conversion efficiency (CV%—Coefficient of variation) [24].

Acoustic Parameter	Ash (CV%)	Walnut (CV%)
*f*_1_ [s^−1^]	923 (0.1)	748 (0.1)
*tanδ*	0.008 (7.9)	0.011 (11.4)
*E/ρ* [GPa]	14.93 (4.5)	9.79 (6.7)
*K* [*m*^4^ s^−1^ kg^−1^]	4.81 (9.8)	5.04 (12.2)
*ACE* [*m*^4^ s^−1^ kg^−1^]	261 (8.5)	246 (10.6)
*RACE* [km s^−1^]	209 (7.7)	153 (9.4)

**Table 2 materials-14-05281-t002:** Moduli of elasticity in longitudinal (*E_L_*), radial (*E_R_*) and tangential direction (*E_T_*) and shear moduli (*G_LR_*, *G_LT_*, *G_RT_*) of ash and walnut wood, used for body of tested electric guitars (CV%—Coefficient of variation).

Material	*E_L_* (GPa)	*E_R_* (GPa)	*E_T_* (GPa)	*G_LR_* (GPa)	*G_LT_* (GPa)	*G_RT_* (GPa)
Ash (A)	17.22	2.13	1.91	1.62	1.13	0.63
CV%	16.6	9.2	8.4	11.3	15.6	14.7
Walnut (W)	8.63	1.34	1.89	1.79	1.10	0.44
CV%	15.4	7.9	5.6	8.9	11.5	12.4

**Table 3 materials-14-05281-t003:** Modal frequencies in Hz of electric guitar with body from ash—(A) and walnut wood (W) (NA—8th fret of the fingerboard; BP—body pickup; M_1_—body microphone; M_2_—edge body microphone; CV%—Coefficient of variation).

	Ash (A)				Walnut (W)			
	NA	BP	M_1_	M_2_	NA	BP	M_1_	M_2_
1st mode	119.8	119.0	119.1	114.1	104.2	108.2	109.2	111.2
CV%	10.2	7.9	8.3	8.0	12.6	12.6	11.7	6.3
2nd mode	201.8	204.7	205.6	200.5	200.1	200.5	201.7	200.3
CV%	0.8	0.4	0.3	0.4	1.8	0.3	5.2	0.4
3rd mode	437.0	440.4	437.6	437.0	418.9	420.6	418.9	418.5
CV%	0.1	1.0	0.1	0.1	0.1	6.1	0.1	0.1

## Data Availability

The data presented in this study are not publicly available due to non-disclosure agreement.

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
