# Peer review of "Comparison of the Vibration Damping of the Wood Species Used for the Body of an Electric Guitar on the Vibration Response of Open-Strings"

_materials, 2021, doi:10.3390/ma14185281_

Round 1
Reviewer 1 Report
The research results presented in the article are interesting. Paper needs to be further improved in order to be recommended for publication. The paper needs to be carefully revised to improve terminology and readability.
Comment 1:
Why was wood from two wood species approximately the same density and structure used in the experiment?
Comment 2:
Subtitle ``2.1 Construction and assembling of electric guitar `` can be omitted.
Comment 3:
On line 75 and 92: the authors should provide more information, i.e. density of wood, moisture content of wood after wood conditioning. These wood properties are very important.
Comment 4:
Can you describe equipment and measurement methodology used at measurement acoustic parameters presented in Table 1?
Comment 5:
Table 1. Unit of frequency fn according to SI is Hz.
Symbols for physical quantities are printed in italics.
Comment 6:
Table 3. add unit below line, i.e.
NA BP M1 ……
(Hz)
Comment 7:
How many times was performed measurement of modal frequencies of electric guitar with body from ash and walnut?
Author Response
Dear referee,
thank you for the review of the manuscript. Please find attach the respond letter with the changes made in the manuscript.

Reviewer 2 Report
This paper compares Ash vs Walnut wood used for the body material in an “electric guitar” by examining the response of the first 3 body modes in terms of frequency and damping. The authors also compare the signal response of the guitar’s 6 strings when fitted, tuned and plucked on Ash vs Walnut bodies.
While an interesting premise (especially for electric guitarists) and a suitably simple and straightforward approach is used to compare the response of Ash vs Walnut bodies in electric guitars, the study in its current form is somewhat limited in scope and scientific merit, and more seriously, suffers from several difficulties in various areas, outlined as follows:
Title:
The study title claims to report rather broadly “Influence of the material properties of the body of an electric guitar on its acoustic behavior”, but disappointingly limits itself to focus only on one material property of wood (vibration damping) and one output acoustic property (damping coefficient of 7 vibration modes of the open strings). The present title is vague/misleading, and must be changed. (Perhaps “Comparison of vibration damping of two wood choices used on the body of an electric guitar on the vibration response of open-strings” or something similarly specific.)
Next, there are several gaps/deficiencies in the methodology used, and it is compounded further with how the collected results are treated, interpreted and discussed. Some examples of issues are as follows:
Methodology:
- Methodology: only one example/sample of each wood body is used? Is this sufficient to convince readers the results are meaningful and representative of the two material/wood species? If so, a clear reasoning must be provided for such a limited approach chosen.
- Wood density and Mass are important factors of guitar material choice, and are in fact introduced as such (Lines 61, 64 etc) but is then curiously omitted from further discussions of material properties (e.g. Table 1 and Table 2) and subsequently in the rest of the paper (“heavier woods” is mentioned only in passing in Line 278). Density and Mass are important factors in mechanical admittance and impedance matching, and must be included in Table 1/2 and subsequently addressed alongside discussions of damping and stiffness for a true/fair picture of material choice to arise.
- The decision of using rectangular blocks of wood for comparison, instead of more conventional electric-guitar body shapes is not explained. Also lacking is any explanation/discussion of how the rectangular body’s response meaningfully relate (or not) to the response of actual conventional electric-guitar bodies.
- The study makes no mention of how the electric-guitar, cradled against player’s body and the player’s hands, straps, etc. contribute to additional damping and influence the vibration response of the string. Would the reported differences in damping between Walnut and Ash become negligible when cradled (which would have rather greater damping + mechanical coupling with the player’s body) during playing? These must be addressed to enable the findings reported here to be appreciated satisfactorily.
- Section 2.2: related to the point above, how is the guitar suspended/rested during impulse excitation and excitation by guitarist? Where are the resting/suspension points? Upon what structures, and what damping is provided to isolate the guitar during excitation? What contact locations are used? These all influence the resulting behavior of the system under excitation and must be disclosed and discussed.
- Line 92-93: The same hard maple composite neck is coupled to both wood bodies. When coupled, force and vibration interaction between body+neck arise due to the different wood properties (body modes, damping, mass, stiffness, etc), which complicate the resulting damping and body-mode analysis (not to mention the presence and mass of the bolts!) and further string vibration mode damping etc. I.e. how can the results reported here be isolated and attributed only to the body material alone? (As an alternative – if neck+body was homogeneously Ash vs Walnut, then any differences reported become unambiguously due to the material choice)
- No description of the strings used are given: material, linear density, gauge, presence/tightness of windings, surface texture, coatings, brand/model etc. String parameters are known to interact at both the bridge and nut/frets (impedance/admittance matching, etc), and influence the resulting damping, mode and radiation behavior etc. (Also, what material and properties are nut/frets and bridge?)
- String naming convention: using “E2” to refer to the lower E-string and “E1” for the higher E-string is not helpful and confusing, not to mention counter-intuitive (why is the first/lower E string numbered 2?). In fact, later in Figure 6 the authors are themselves confused, and erroneously refer to E1 in the caption when the Figure indicates E2. To avoid these confusion and improve clarity, the string names introduced in Lines 97-98 should instead conform to the more commonly used and unambiguous Scientific Pitch Notation (https://en.wikipedia.org/wiki/Scientific_pitch_notation) as E2, A2, D3, G3, B4 and E4 (or some similarly unambiguous format)
- Line 100: “directional mechanical stiffness” is mentioned, but the reader is not informed of the grain direction of the Ash and Walnut used in the rectangular bodies; also, given the two species have different grain structure, how can “directional mechanical stiffness” be considered to be equivalent, to achieve a fair comparison?
- Section 2.2.1: how are locations for EP, NP, M1, BP and M2 determined? How do these positions avoid anti-nodes? Or, how do these positions maximize representation of the modes of interest?
- Why is the Impulse Excitation Point at the guitar neck (and why 8th fret)? Disclose the rationale. Further, if the purpose of the impulse excitation is to study the body wood, wouldn’t it make more sense to position the Excitation Point directly on the body instead?
- Section 2.2.2: “Excited by hand” – how? With a pick? With nails? Plucked 45° into the body? Or away from the body? A photo (close-up, with annotation) will be useful. How is the baseline intensity (loudness/voltage) level determined to allow meaningful comparison across the 6 strings which have different mass, tension and frequencies? Is there any indication of the typical deflection distance before release or the resulting dB level measured? This would be helpful for readers to understand better how reproducibility is indeed satisfactory (line 157)
Results and Discussion:
- What are the body vibration modes 1, 2 and 3 in question? A schematic would be most helpful to the readers – especially for understanding why mode 2 seems to be least affected by material choice.
- Table 3 presents the modal frequencies determined for the first 3 modes for both Ash and Walnut, where Walnut has slightly lower values for 1st and 3rd mode (~10 Hz and ~19 Hz respectively), but remains similar for 2nd mode (~3 Hz). However, it is not clear what the significance of the similarities and differences between these values are (is 10-19 Hz deviation meaningful?). Also, we know that modal frequencies arise from both stiffness and mass, but no mass information is provided – the analysis and discussion (see line 260-263) regarding the results of Table 3 is incomplete. Importantly, what is the significance of the higher (or lower) modal frequencies of the body on the output performance of the guitar? Bear in mind these modes are relatively low values (<500 Hz) with regards to the string excitation frequencies, whose rich partials rise well above 500 Hz; how will these differences implicate musical/performance/perception considerations? (E.g, the human ear is most sensitive between 2-5 kHz, so modest changes in the coupling of low-frequency partials – sounding alongside rich upper harmonics – may arguably not be very perceptible). A discussion on the significance of Table 3 is lacking and must be included.
- Figure 4: Vibration Damping [absolute] values measured using M1 and BP are not in agreement, and not addressed – why? An explanation should be offered.
- Four simultaneous measurements were made (M1, M2, NA and BP) but Figure 4 shows only data for M1 and BP – why were M2 and NA omitted? (I understand NA shows the neck response, but it would also be helpful to see their damping values alongside M1 and BP, since the same neck is used for both Ash and Walnut cases)
- Figure 4 & 5: An explanation (even speculative) should be offered for why Modes 1 and 3 are different for Ash and Walnut while Mode 2 is similar – both in terms of modal frequency and vibration damping. Likewise, why does Figure 5 show stronger correlation between M1 and NA for Modes 1 and 3?
- Figure 5: What are the implications of a strong (or weak) correlation in damping? What material property would affect modes 1, 2 and 3 differently, and why? A discussion/explanation (even speculative) is missing and must be supplied in order to make sense of Figure 5 satisfactorily.
- Figures 6 and 7 show the damping coefficients for the first 7 vibration modes of the 6 open strings, and line 236-237 states “the damping coefficient was generally highest at the fundamental frequency”. This is somewhat counter-intuitive – the fundamental partial in a plucked string has typically the longest decay time, lasting well after the higher partials have dissipated; decay time and damping coefficients are inversely correlated, yes (cf. Lines 284-286 and 298-299)? This would imply that we should instead expect the Damping Coefficient plots to rise with mode number? (Unless I am mistaken – apologies if so; perhaps there is something with how the coefficient is defined here?)
- No attempt is made to offer the reader any explanation for why E2 and A2 (and maybe D3) seem more sensitive to body material, but not the other strings.
- Is the string damping coefficient influenced only by the body? What about the effect of string’s construction and material? (e.g. presence of winding? The wound first 4 strings seem to have more similar behavior, while the unwound B3 and E4 strings are more mutually similar)
- How are open string measurements indicative of fretted notes played? What about non-open string notes? A discussion must be made.
- Line 262 “…we achieve higher modal frequencies…” This statement begs a discussion of how a 10-19 Hz difference is meaningful? And again, any discussion of stiffness and modal frequency is not complete without also considering density/mass.
- Line 275 mentions “it is necessary to choose a wood with favorable damping properties” – what is “favorable” and who gets to decide? In an electrically amplified instrument, is more or less damping more desirable? How so?
Conclusions:
- Line 297 “…we prefer ash wood to the less stiff walnut, the latter having a higher damping of vibrations.” Stiffness and Damping are not necessarily correlated, so the statement is not intuitive. Also, the statement ignores the other important parameter of density/mass.
- Line 300-302: This concluding sentence of the report makes little semantic sense at several levels. You have not demonstrated how natural frequencies of the guitar should have anything to do with the string: why bring that up right at the conclusion? Does the 10-19 Hz difference really matter to the output string behavior? Further, what has body damping got to do with the string and guitar frequencies? Is this something you have already demonstrated?
Sloppy Writing:
While most of the written text is reasonable for a scientific report, instances of casual (sloppy) writing arise, with errors in Figure Captions (obvious copy-and-paste), abbreviations in main body text, etc, which are easily avoidable if the authors were more careful. Examples:
- Figure 8 (line 254) is mislabeled as Figure 7!
- Figure 3 Caption (line 187) mentions “CV% - Coef. of variation” when there is no such data presented in Figure 3; instead, the string of text seems to have been copied-and-pasted from the Table 3 Caption.
- Use of abbreviations when the full word should be used, E.g. Lines 181, 187 “Coef.” when “Coefficient” should be fully spelt out; Line 195 “stat. significantly higher” should be “statistically”? Check if this (or similar) occurs elsewhere too.
- Lines 94-96: “pickups DiMarzio DP103CR 94 36th Anniversary, PAF (near the neck) and DiMarzio DP223BC PAF 36th Anniversary 95 (near the bridge)” What is “PAF”? Are the pickups in fact the same model?? If so, why are they named differently each time and repeated separately? Similar frustrating inconsistency at Lines 116 and 159 (they are no longer "36th Anniversary"?).
- Standard practice: all physical quantities should be italicized, e.g. “f1” should be “f1”
Non-factual/Sweeping Statements:
There are some broad statements, which are untrue if presented without further qualifying/clarification. Care should be taken to accurately convey these observations and avoid overreach:
- Line 33 “the electric guitar itself does not radiate sound” – while the radiated signal may indeed be minimal when compared with classical or folk guitars with a hollow body, the solid body of the electric guitar does indeed still transmit some acoustic output to the air, as indicated by the use of Microphones M1 and M2 in the methodology. Please check for similarly sweeping statements elsewhere in the paper and make the necessary amendments.
- Line 60-61: I’m pretty sure many owners and performers of acoustic guitars would disagree that their wood quality is any more inferior (or better) than electric guitars – this statement is unnecessary at best, and certainly spurious.
- Line 102: how can the headstock and body of the guitars be “practically symmetrical” when the strings loaded have different mass and tensions? The resulting vibrations will not be symmetrical even if the geometry might be. (Also: symmetrical with respect to which axis?) What are the implications in this study for symmetry (or lack thereof)? (A conventional electric guitar body is certainly not symmetric along its long axis, nor is the source of force interaction on each string.) Is the symmetry worth mentioning?
- Figure 6: the authors erroneously refer to E1 in the caption when the Figure (and context) indicates E2
- Line 238: “The study confirmed no difference in damping coefficient…”. This is false – there is a difference, and the differences (where they arise), are no smaller than differences seen in D-string in Figure 6.
- Likewise Line 250: “…showed no difference…” Be careful - there are clear visual differences, although perhaps not as stark as for E2 and A2
- Line 259: “…they significantly affect…” – Really? A few Hertz difference in resonance frequencies and some (inconsistent) shifts in damping coefficient = significantly affect?
Author Response
Dear Reviewer,
thank you very much for the revision of the manuscript. Please find attached the document with our response.

Reviewer 3 Report
This study explores the influence of the wood type of the body of a solid body electric guitar.
It concludes that a wood with higher stiffness and lower damping will ensure longer sustain
for the guitar, which is an interesting property for players.
This conclusion is in line with previously existing studies on the topic, but do not
really bring novelty.
What is claimed to be a novelty in the abstract, inlcuding the string in the measurement,
have in fact not been analyzed, and no material is provided to investigate this issue.
The material presented are to few to addequately support the conclusions: the measurement
have been performed only at one location which do not correspond to the contact point of
the open strings, whose plucked behavior have been analysed. Thus no comparison
between the measured mechanical properties and the plucked behavior is possible.
The previous studies on the topic have relied on measurement at multiple location and
multiple playing position, not only open strings.
So the measurements should be extended to have a more global view of the mechanical
behavior of the instruments, deepen the analysis and support more efficiently the conclusions.
The phenomenon of dead tones reported in previous studies is not discussed here, and
as a mater of fact, the limited measurement do not allow that.
Detailed comments:
******************
Abstract:
It is said that one of the novelties of this study is to take into account the coupling
between the body of the guitar and the strings. One expect that it is said in the abstract what
this changes compared to other studies. Also such a discussion do not appear in the body of the
article. So this should be included, or removed from the abstract if no work in this direction
have been done. The material presented on the article do not allow to conclude on the effect
of coupling between body and strings: there are no measurement without strings to compare
to the measurements presented. So if this question is to be investigated new measurement
should be made.
Introduction:
paragraph 2: it would be more accurate to say that solid body guitars are not designed to radiate
sound: they radiate sound, even though it is weak, as a matter of fact musicians sometimes
practice with the instrument unplugged.
paragraph 3: Paramount is a bit exagerated
Methods:
section 2.1
paragraph 1: explain to what correspond the different measured parameters of Table 1
These relevant values of these parameters should be explicitely and specifically refered to
in the discussions when the influence of the mechanical properties are discussed.
Table 2: where does these data come from? Did you measure it? Or is it from a database ?
Precise.
section 2.2.1
Only one measurement point is very few: the admittance can vary a lot depending on the
location on the neck. Dead tones are produced at specific location and specific frequencies,
and thus cannot be spoted with just one measurement.
Another problem is that the contact point of the played strings are not the same as the
hammer impact point, so no comparison can be made between the two type of measurement.
Precise if the string are damped during hammer measurement?
On figure 2 it seems that damping material is placed under the strings to absorb their
vibrations. If this is the case, this could affect the coupling of the string with the body
and it should be discussed.
section 2.2.2
time-frequency sonograms --> spectrograms
section 3.1
From what I understand the signal of the accelerometer is used directly, and no transfer
function between the hammer and the accelerometer is computed. This could be easily done
and would ensure cleaner and more relevant measurements.
Also, it would be easy to estimate the neck local admittance. This may be further improved
by moving the accelerometer on the other side of the neck closer to the impact point.
What does mean statistically significant?
Precise what statistical analysis you have used, and what value support your claim.
It appears that you relied only on correlation coefficient measurements. This indicate
a correlation but not necessarily a causal relationship. So the potential mechanismes
explaining the observed correlations must be explained.
section 3.2
decay time for open string: the end of the string is at a different position than the
excitation with hammer. So it is not possible to compare. To compare one should have the same
excitation point.
4 Discussion
1st paragraph
acousto-mechanical --> mechanical
Was the body and the whole instrument measured separately?
It seems that there is a misunderstanding about the vibratory properties of the neck:
it makes not that much sens to discuss the separate properties of the neck and body from an
observation of the whole instrument. It is obvious that the whole instrument have different
properties than its separate parts alone.
Also in this discussion could be added the effect of the presence of the string if a dedicated
experiment is done.
2d paragraph
Sentence "Regardless, the part..." --> split in two, also sentence is obvious: admittance is a vibromechanical property.
What showed the previous studies is that the problem is a bit more complex than that:
if one want to avoid dead tones, one has to consider the vibration modes of the guitar.
Also these conclusions are a bit obvious, and it is not necessary to do this kind of study to
come with these conclusions.
Author Response
Dear referee,
thank you very much for the review of the manuscript. Please find attached the respond letter.
